# Graphene Oxide as a Nanocarrier for Biochemical Molecules: Current Understanding and Trends

**Soumajit Mukherjee [1], Zuzana Bytesnikova [1], Amirmansoor Ashrafi [1], Vojtech Adam [1,2] and Lukas Richtera [1,2,\*]**

[1] Department of Chemistry and Biochemistry, Mendel University in Brno, Zemedelska 1, CZ-613 00 Brno, Czech Republic; soumajit.mukherjee@mendelu.cz (S.M.); zuzana.bytesnikova@mendelu.cz (Z.B.); amirmansoor.ashrafi@mendelu.cz (A.A.); vojtech.adam@mendelu.cz (V.A.)

[2] Central European Institute of Technology, Brno University of Technology, Purkynova 123, CZ-612 00 Brno, Czech Republic

\* Correspondence: lukas.richtera@mendelu.cz; Tel.: +42-054-513-3311; Fax: +42-054-521-2044

**Abstract:** The development of an advanced and efficient drug delivery system with significant improvement in its efficacy and enhanced therapeutic value is one of the critical challenges in modern medicinal biology. The integration of nanomaterial science with molecular and cellular biology has helped in the advancement and development of novel drug delivery nanocarrier systems with precision and decreased side effects. The design and synthesis of nanocarriers using graphene oxide (GO) have been rapidly growing over the past few years. Due to its remarkable physicochemical properties, GO has been extensively used in efforts to construct nanocarriers with high specificity, selectivity, and biocompatibility, and low cytotoxicity. The focus of this review is to summarize and address recent uses of GO-based nanocarriers and the improvements as efficient drug delivery systems. We briefly describe the concepts and challenges associated with nanocarrier systems followed by providing critical examples of GO-based delivery of drug molecules and genes. Finally, the review delivers brief conclusions on the current understanding and prospects of nanocarrier delivery systems.

**Keywords:** graphene oxide; graphene; nanocarrier; drug delivery; nanomaterials; cancer drug; gene delivery

## 1. Introduction

Parallel to the complexity of drug design, one of the key challenges facing tissue-specific targeted treatment in the healthcare system and in vivo scientific experiments is the delivery of biomolecules or drugs to the correct compartment or tissue of interest [1,2]. Some of the technological challenges include the poor solubility of drugs in biological media, protecting biomolecules from various enzymatic attacks, and reaching the target by crossing the semipermeable cell membrane and further compartments, such as endosomes. [3–6]. Cell membranes are selective in transporting molecules inside and outside of the cell [7]. Designing the appropriate molecular structures and controlling the surface ionic charge for interactions with cell surface ligands or channels have allowed the delivery of clinical drugs to cells via the active or passive mode of transport. For the past few decades, biologists have been trying to formulate new types of biochemical solutions, such as precursor-based, DNA-based, or protein-based therapeutics, to improve targeted delivery. The major advantage of this approach has been the ability to avoid enzymatic degradation of biomolecules. Proper synthesis of a drug in vivo from ectopically

expressed genetic material following molecular folding is often impossible to achieve under various biological conditions.

By harnessing nanoscience and chemical synthesis knowledge, experiments with novel nanomaterials toward achieving better medicinal efficacies have been possible. The abundance of low-cost nanomaterials in various engineering and healthcare sectors has led researchers to utilize their favorable properties. Advancements in nanotechnology in combination with synthetic chemistry have helped in formularizing new drug delivery systems using nanomaterials [8,9]. Targeted and tissue-specific drug delivery systems are advantageous over conventional systems. These systems not only ensure the delivery of the drug but also reduce toxicity and increase therapeutic efficiency [10,11]. By description, a nanocarrier is a nanomaterial vector or carriage system used to transport another substance, most often a drug molecule [12,13]. Advanced chemical synthesis has enabled researchers to produce controllable and tunable nanomaterials differing in size and shape and with diverse intrinsic and exterior properties suitable for different requirements. In recent years, nanocarriers have demonstrated tremendous potential in drug delivery associated with difficult tissue locations or complex disorders [14–17]. Nanocarriers have also been studied as competitive substitutes for conventional chemotherapy practices to achieve tissue-specific delivery and reduce harm to normal cells. Nanocarriers are tailor-made to provide a defensive cover to a drug or biomolecule to protect it from detrimental environments (extreme pH and light) or biological factors (enzymes) [18]. Additionally, nanocarriers can be tuned for controlled and slow drug release [19]. In addition to conventionally synthesized drugs, nanocarriers have been used to carry mature or precursor protein molecules, therapeutic peptides, antibodies, and small noncoding RNAs with therapeutic activity. Several critical attributes define an ideal nanocarrier suited for most biological and medical applications, such as size, shape, net charge, carrying capacity, lack of immunogenicity, and clearance from the biological system [20].

Liposomes, micelle-based nanomaterials, polymers, and organic carbon-based nanomaterials are some of the most well-studied and commonly used nanocarriers in recent times [21–25]. Recent advancements in the use of polysaccharide bionanocomposites along with graphene derivatives in drug therapeutic and pharmacogenomic applications have been reviewed [26]. Nanomaterials are well suited for applications as nanocarriers due to their nanoscale size, inertness, and biocompatibility. The biomedical field has witnessed a surge in applications of graphene-based nanomaterials, including new-generation, advanced drug delivery approaches, due to their many interesting physicochemical properties, such as their large surface area with a 2D planar structure, thermal and chemical stability, biocompatibility, and ability to bind and carry molecules on the surface. The large surface area and presence of favorable functional groups for future modification make graphene oxide (GO) an interesting choice as a nanocarrier for biochemical molecules. Additionally, the π-conjugated structure of GO is advantageous for capturing drug molecules through π-π stacking interactions or efficient covalent bonding. Technically, the physical properties of GO are dependent on the particular method of synthesis and degree of oxidation. Efficient cellular internalization is one of the rate-limiting steps in the development of an efficient drug delivery formula. In a very recent study, the authors described the production of nanographene oxide (NGO) particles via mild oxidation of graphene materials that exhibited massive cellular uptake [27]. Depending on the method of GO chemical synthesis, many functional groups, such as carbonyl, carboxyl, hydroxyl, and epoxy groups, can be found for covalent interactions with biomolecules. In addition to being used in cancer treatment, GO-based nanocarriers are currently being explored for the treatment of many complex disorders, such as Alzheimer's disease, Parkinson's disease, and inflammatory bowel disease, as well as to treat viral infections [14–17]. The goal of this review is to summarize and address the most recent uses of GO-based nanocarriers and related improvements in drug delivery.

## 2. Critical Attributes of Nanocarriers for Drug Delivery

The conventional mode of transportation for chemotherapeutic agents poses several critical challenges, such as their toxicity to healthy cells. Poor selectivity results in many side effects, encapsulation of drug–carrier conjugates in biological compartments, and less specificity, which altogether significantly reduce drug efficiency [28,29]. In practice, to reach their target, most drugs must pass through the selectively permeable cell membrane, which is a protective wall separating the cytoplasm from the external environment. Although specific binding with the target is crucial for drug potency, failure to reach the target due to poor permeability often translates to poor or nonexistent in vivo drug efficacy. Nanocarrier-based platforms are colloidal systems with submicron-sized nanoparticles (NPs) of typically less than 500 nm, which are also characterized by a large surface-area-to-volume ratio [30]. In vivo, NP-biomolecule conjugates with a size of less than 10 nm also face the risk of being eliminated from the system by renal clearance [31]. Many studies have found that cellular uptake or efficiency of internalization is highly dependent on size (reviewed in [32]) (Table 1). A large surface is a particularly essential property to improve the carrying capacity of a carrier. The electrical charge of the conjugate surface also plays an important role in ensuring proper cell permeability (Figure 1). Specialized proteins or channels are required for charged particles or ions to cross the membrane (reviewed in [33]). Studies suggest that slight changes in size and surface charge often have significant effects on the cellular uptake of NPs [34]. Some of the other challenges for any drug delivery system are biocompatibility and predicting the appropriate interaction between a synthetic drug molecule and a biological ligand.

**Table 1.** Comprehensive list of widely studied nanoparticles (NPs) and their internalization efficiency based on size.

| NP Material | Size (nm) | Major Observation | Cell Type | Ref. |
|---|---|---|---|---|
| Au | 2–15 | Smaller NPs have better mobility; 2–6 nm NPs localize in the cytoplasm and nucleus, 15 nm NPs localize in the cytoplasm only | MCF-7 | [35] |
| | 2.4–89 | 2.4 nm particles were found in the nucleus and 5.5–8.2 nm particles in the cytoplasm; no cellular uptake was observed for NPs larger than 16 nm | Cos 1 | [36] |
| | 2–100 | 40–50 nm NPs exhibited the best cellular uptake | SK-BR-3 | [37] |
| | 4–17 | Cellular uptake efficiency was enhanced with an increase in the size of the NPs | HeLa | [38] |
| | 14–100 | Maximum uptake was recorded at 50 nm | HeLa | [39] |
| | 45–110 | Maximum efficiency was reached with 45 nm NPs | HeLa, CL1-0 | [40] |
| QDs | 4–7 | Cellular uptake efficiency was size-dependent | A-427 | [41] |
| $Fe_3O_4$ | 8–65 | The highest uptake efficiency was recorded for 37 nm NPs | RAW264.7 | [42] |
| Polystyrene | 20–100 | The fastest cellular entry was observed with 40 nm NPs | A549 | [43] |
| | 40–2000 | Cellular uptake was highly size-dependent; slow migration was observed for larger NPs | HeLa, A549 | [44] |

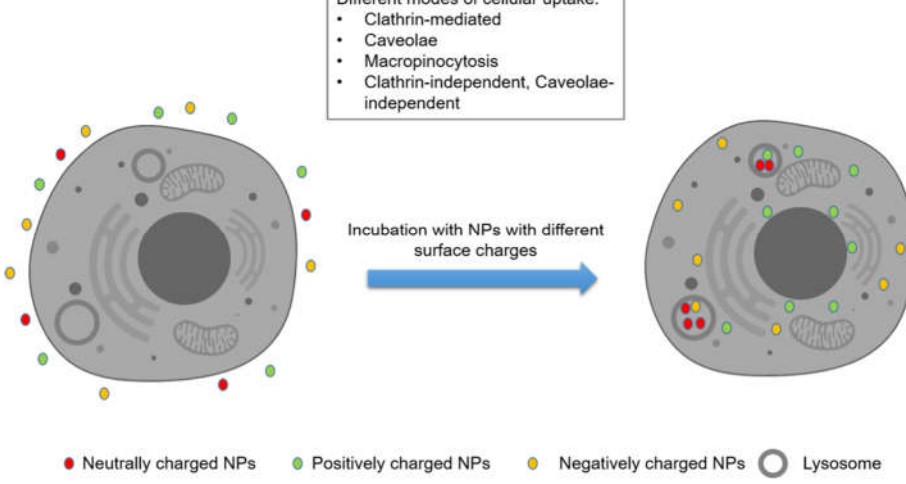

**Figure 1.** Cellular uptake of NPs with different surface charges.

### 3. Concept and Challenges of GO-Based Nanocarriers for Drug Delivery

In the past few decades, various nanomaterials with different characteristics in terms of size, shape, and composition have been studied in drug delivery attempts, such as various metals and semimetals, carbon nanotubes, liposomes, and micelles. Graphene and its derivatives possess advantageous properties and thus have emerged as a promising solution for systematic, target-specific, and controlled release of biomolecules in vivo. The planar 2D structure of graphene is well suited for binding and immobilizing various substances, such as synthetic drugs, metals, biomolecules, and fluorescent probes (Figure 2). Compared to carbon nanotubes (CNTs), graphene exhibits some crucial qualities, such as economically easy synthesis, a large surface area, and greater biocompatibility, due to the significantly lower amount of toxic metal residues.

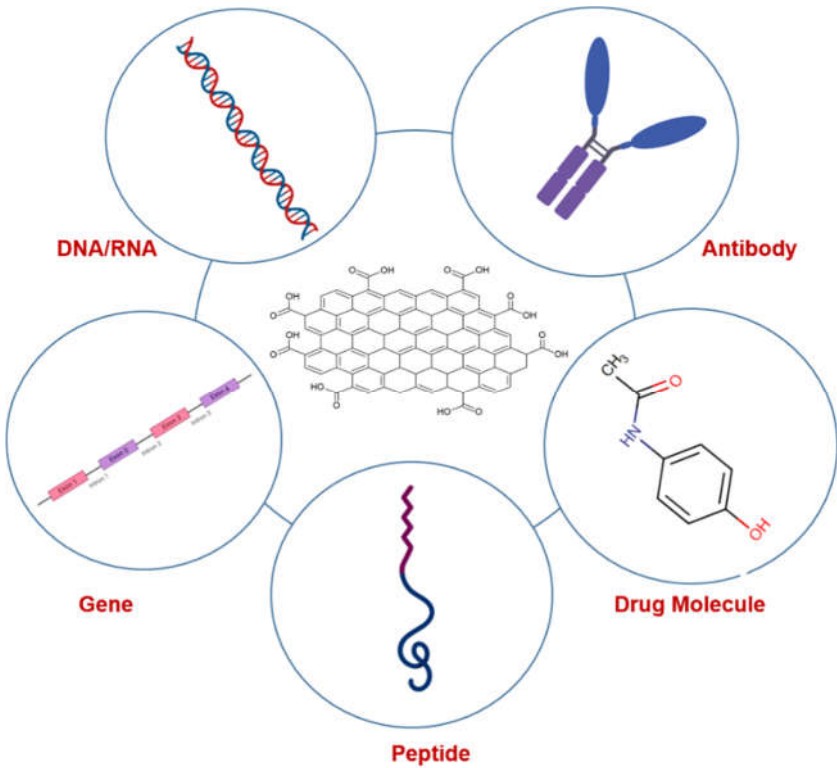

**Figure 2.** Illustration of common modifications of graphene oxide (GO) as a nanocarrier.

Several factors often play important roles in the potency of an application, including surface area, surface chemistry, number of layers, and purity. Graphene has a theoretical surface area of 2.630 m$^2$/g, which is almost three times higher than that of CNTs and much higher than that of other nanomaterials tested for drug delivery. It has been reported that the loading ratio of graphene-based nanocarriers reaches as high as 200% compared to other contemporary NPs and drug delivery systems [45]. In its atomic-scale hexagonal pattern, carbon atoms are densely packed and exposed on the surface of a graphene monolayer, which significantly promotes the drug-binding capacity on the surface compared to other nanocarriers. High lateral dimensions and a large number of layers can impose size limitations that affect uptake, crossing of biological membranes, renal clearance, and other biological interactions, especially after an overall increase in size after conjugation. According to previous studies, the shape of an NP significantly impacts cellular internalization. Graphene-based nanomaterials are uniquely designed in a planar two-dimensional morphology, unlike other biomolecules, and have advantages over tubular (CNT) or spherical (such as gold) nanomaterials. The thickness and number of layers in GO are important parameters for several reasons. Structural rigidity is often required while interacting with a cellular ligand or for cell permeability. A large number of layers in GO reduces the surface area but provides rigidity to the structure for efficient cellular entry, but if GO is too rigid, the conjugate can damage the cellular

structure, which will impact the efficiency of the drug delivery mechanism. Therefore, the ability to control and maintain an optimal rigidity of the material could be a challenge in designing an efficient drug delivery system.

The synthesis method for graphene-based nanocarriers and thus the exact chemistry of the output material has a significant impact on the efficiency of the system. The surface chemistry of pure graphene makes it highly hydrophobic and poorly dispersible in water, which makes it a difficult material for in vivo use. The presence of a surfactant or further surface modification is required to relatively increase the solubility of graphene. In contrast, GO is hydrophilic due to having surface functional groups and is an economically suitable substitute for pristine graphene. However, depending on the GO production route, various impurities and residues can be present, such as permanganates, peroxides, nitrates, sulfates, and some oxidative debris, which negatively impact the purity of the nanocarrier and biological interactions and can influence toxicity to cells [46,47]. Although time-consuming, rigorous washing of the nanomaterials after synthesis plays a crucial role in determining the purity and removing both inorganic and organic contaminants.

In a broad sense, designing a drug delivery system with graphene and GO-based nanocarriers is challenging and will have to successfully address the above discussed physical and chemical attributes. To summarize, the most crucial steps of drug delivery systems are to decide the type of modifications required depending on the load or the chemical structure of the therapeutic agent, followed by improvement of the biocompatibility and control of toxicity. Finally, the drug delivery system should be engineered for efficient biodistribution and controlled release of biochemical molecules. Significant advancements involving in vivo studies have already been reported for graphene and GO-based drug delivery mechanisms, which will be discussed in the following chapters.

## 4. Surface Functionalization and Modification of GO

To develop a drug delivery system with enhanced biocompatibility and the ability to tune the slow release of the drug, the surface chemistry of a nanocarrier is a key aspect. As mentioned earlier, pristine graphene is highly hydrophobic and poorly dispersible in water, and appropriate surface modifications are required to introduce oxygen-containing functional groups to make it dispersible. The NP is the core material of a drug delivery system, and thus, proper formulation of surface modifications is of the utmost importance to ensure the immobilization of biomolecules. Over the years, various methods have been established for surface modifications of graphene and GO using covalent and noncovalent bonding methods (Figure 3). The oxygen-rich functional groups or reduced doping elements on GO can be efficiently treated as catalytically active domains for covalent/noncovalent modifications to suit the requirement of a particular application. Covalent bond modification of graphene and GO essentially involves the introduction of useful functional groups via covalent bonding to improve many properties, such as increased dispersity and biocompatibility, and is possible due to the presence of defects or reactive oxygen groups in the lattice [48]. Covalent modifications can be achieved by several methods, such as electrophilic addition, nucleophilic substitution, condensation, and addition, whereas electrostatic forces, van der Waals interactions, and hydrogen bonding are used to achieve noncovalent modifications (Table 2).

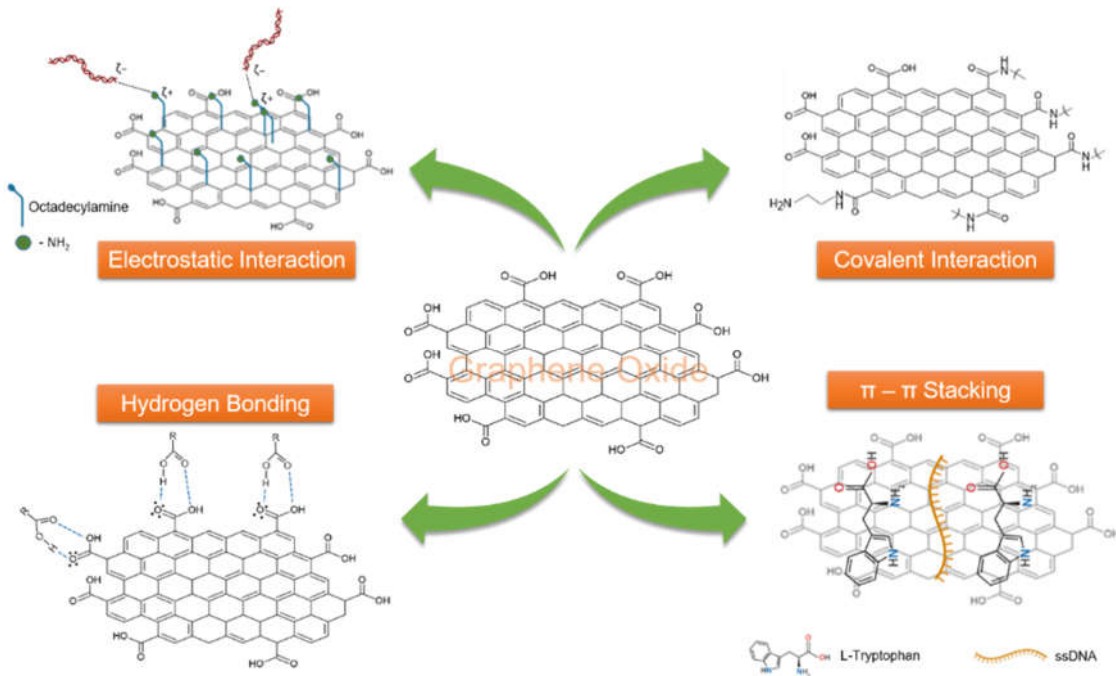

**Figure 3.** Schematic representation of common modifications of graphene oxide.

**Table 2.** Summary of frequently prepared covalently and noncovalently modified GO for drug delivery purposes.

| Type of Bonding | Material or Mechanism | References |
| --- | --- | --- |
| Covalent bonding | Poly(ethylene oxide) | [49–52] |
| | Folic acid | [53,54] |
| | Chitosan | [55–57] |
| | Poly(iminoethylene) | [58,59] |
| | Poly[imino[(2S)-2-amino-1-oxo-1,6-hexanediyl]] | [60] |
| | Iron(III) oxide | [61,62] |
| | Gelatin | [63,64] |
| | Dextran | [65,66] |
| | Poly(vinyl alcohol) | [67] |
| | Sulfonic acid | [68,69] |
| Noncovalent bonding | π-π stacking interaction | [70–72] |
| | Electrostatic bonding | [73–75] |
| | van der Waals force | [76,77] |
| | Hydrogen bonding | [73] |

## 5. Modification of GO as a Nanocarrier with Therapeutic Molecules

In recent years, due to its favorable properties, GO has been substituted for CNTs in efforts to develop novel nanocarrier systems [78]. This section discusses recent advancements in nanocarrier systems and is categorized into subsections depending on the type of therapeutic agent. The addressed research examples greatly increase the value of the core nanomaterial in the sense of developing a precisely targeted drug delivery mechanism with a controlled and stimulated biodistribution mechanism.

### 5.1. Delivery of Drug Molecules

Some of the biggest challenges in the study of novel drug administration are the transport of the molecule at the right concentration to the tissue site of interest, evading enzymatic attack in body fluids, and efficient cellular uptake. Several existing drug carrier systems have shown good efficiencies, such as increased solubility and prolonged circulation time, but the efficacy of such

systems is still insufficient for clinical use, mostly due to the inability to load a high amount of the drug and low surface functionality. Moreover, poor cellular uptake and distribution in nonspecific normal tissues resulting in significant toxicity have restricted the potency of nanocarrier systems. Thus, the development of a nanocarrier system with high carrying capacity, specificity, and biodegradability; low toxicity; and good, controlled drug-release efficiency is still needed.

Toxicity and specificity are two major setbacks in the development of any existing or novel anticancer therapeutic agent. Chemotherapy drugs are notoriously known to exhibit lethal side effects. In recent decades, several micelle-based, polymer, and liposome-based structures have been successfully shown to be able to deliver a drug to the tumor site, driven by many physiological parameters, such as pH and hypoxia. Efforts have been made to successfully integrate GO with proper surface modifications to carry a drug load to tumor sites with substantially better biocompatibility. The immobilization of specific ligands that recognize various molecular receptors or so-called signatures on the tumor cell surface onto a nanocarrier has greatly boosted efforts in making drug delivery target-specific; well-studied target-specific ligands include targeting peptides, folic acid (FA), monoclonal antibodies, and transferrin [79–81].

Relevant studies have shown the efficiency of tumor cell targeting by various ligand-modified polymers, such as mannose-modified cyclodextrin [82]. Galactose is crucial for targeting hepatocellular carcinoma through the asialoglycoprotein receptor. Galactose-modified chitosan (CS) has excellent biocompatibility characteristics for use in drug delivery applications, although its carrying capacity is significantly low. Wang et al. prepared galactosylated chitosan-modified GO (GC-GO), harnessing both the targeting ability of galactose and the high carrying capacity of GO [83]. In the study, doxorubicin (DOX), a widely used anticancer drug for selective killing of B-cell lymphoma, was used as a model drug for delivery. The delivery efficiency was compared with that of CS-modified GO (CS-GO). Cellular uptake and proliferation experiments in HepG2 and SMMC-7221 cells showed that GC-GO-DOX was highly cytotoxic compared with CS-GO-DOX. This observation was consistent with data obtained from in vivo mouse antitumor experiments. Fluorescein isothiocyanate (FITC) labeling was used to show the efficiency of GC-GO-DOX cellular uptake (Figure 4B). In a recent report, a systematic strategy was described in which aminated GO was chemically modified with CS to load DOX [84]. FA was conjugated with CS to enhance target specificity. Drug release from this described CS-GO/DOX nanocarrier system was shown to be pH-dependent. The rate of release was significantly higher in the pH range of approximately 5.3 than in the physiological pH range. The delivery system was shown to be highly efficient in drug loading applications. Recently, Afzal and colleagues demonstrated the use of zinc oxide (ZnO) along with GO to enhance drug loading capacity and control cellular uptake [85]. ZnO increased the targeted drug delivery efficacy because it is biocompatible.

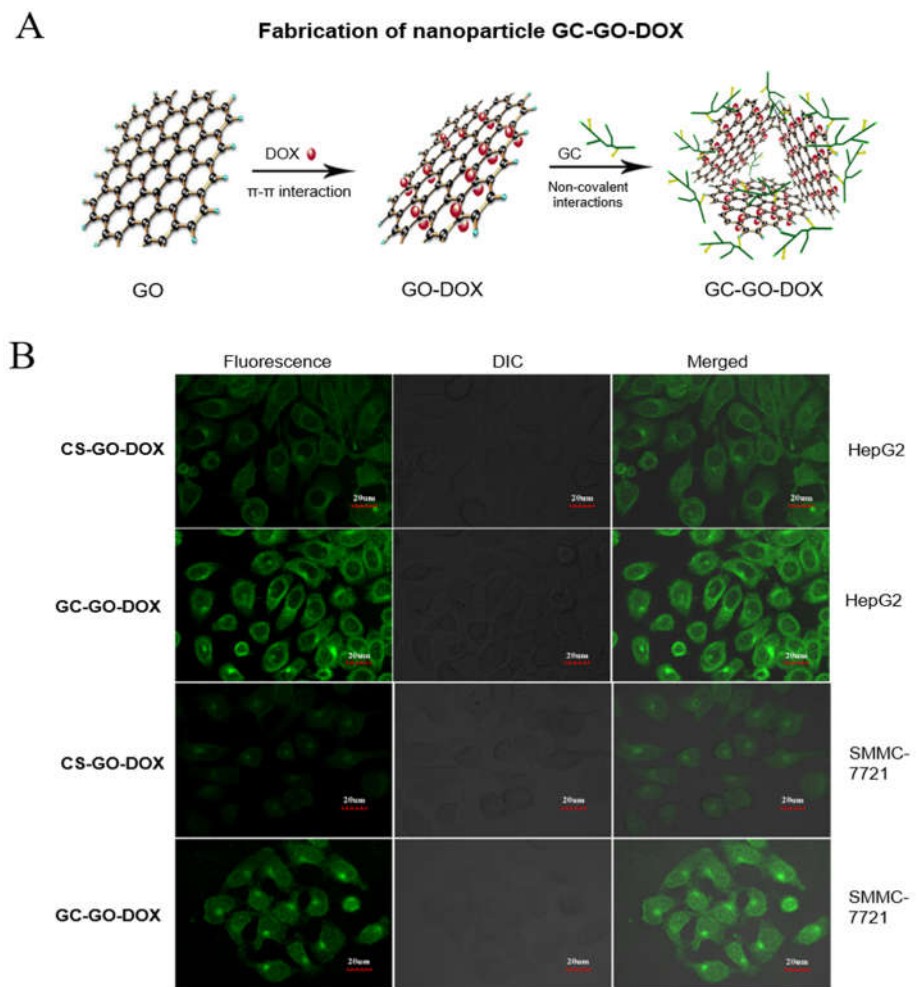

**Figure 4.** (**A**) Schematic illustration of the synthesis of galactosylated chitosan-modified GO-Doxorubicin (GC-GO-DOX) (**B**) localization and intracellular imaging of fluorescein isothiocyanate (FITC)-labeled GC-GO-DOX (green) and chitosan-modified GO-Doxorubicin (green) in HepG2 and SMMC-7221 cells 2 h post-transfection, scale bar 20 μm. (Reproduced with permission from [83]. Copyright 2018, Elsevier).

Dai and colleagues demonstrated the use of poly(oxyethylene)-modified GO, more commonly known as polyethylene glycol-modified (PEGylated) GO, as a drug carrier, which exhibited target-specific delivery [86]. Single layers of NGO were used to covalently bind PEG on the surface, which was further modified with the B-cell-specific anti-CD20 antibody rituxan (NGO-PEG-rituxan). The researchers showed that this conjugate could bind specifically with B-cells in a mixed solution of B-cells and T-cells. Further, this compound was incubated with DOX. NGO-PEG-rituxan-DOX showed significantly higher cytotoxicity against B-cell lymphoma and demonstrated the efficiency of GO as a vehicle for drug delivery. In another study, the same research group immobilized SN38, an analog of an alkaloid with anticancer effects named camptothecin (CPT), onto NGO-PEG via a noncovalent modification and compared its cytotoxicity against an SN38 prodrug named irinotecan (CPT-11) in the human colon cancer cell line HCT-116 using an in vitro cytotoxicity assay [87]. The researchers found that NGO-PEG-SN38 had significantly higher cytotoxicity than CPT-11. In another study, Yang et al. employed a noncovalently bonded GO-DOX nanocarrier system and evaluated the controlled release of DOX in different pH environments [88]. Notably, in the mentioned work, the researchers demonstrated that the carrying capacity of GO was 200% compared with other NPs tested for drug delivery, where the carrying capacity is usually less than 100%. In another effort to develop dual-drug combination therapy, Pei et al. reported the use of PEGylated GO (pGO) and two anticancer drugs, cisplatin (Pt) and DOX, in a conjugate formulation

for targeted delivery and inhibition of tumor progression in vitro and in vivo [31]. Pt is a well-studied cytostatic complex compound that has been used as a therapeutic drug to treat different tumors, such as urinary bladder and neck cancer [89,90]. As a typical Lewis acid, Pt was bound to pGO by the formation of coordinate covalent bonds to prepare Pt-pGO, and then, DOX was immobilized via noncovalent π-π interactions to prepare the dual-drug delivery system Pt-pGO-DOX. Target-specific delivery of the dual-drug system was examined using fluorescence microscopy, whereas an in vitro cytotoxicity assay revealed Pt-pGO-DOX to be highly capable of inducing apoptosis and necrosis while inhibiting tumor growth. The dual-drug conjugate exhibited the highest efficiency compared with pGO-Pt and pGO-DOX, demonstrating the potency of the dual-drug delivery mechanism (Figure 5). Rao et al. described a mechanism of drug delivery using GO modified with a cellulose derivative to increase its loading capacity and improve its release control and biocompatibility [91]. The researchers used carboxymethylcellulose (CMC) to bind with GO (GO-CMC) and loaded a high amount of DOX onto the nanocarrier matrix. The in vitro release of the drug in HeLa cells and NIH-3T3 cells was reported to be pH-dependent. DOX molecules were quickly released in the highly acidic compartment of lysosomes as the conjugate entered the cells via endocytosis.

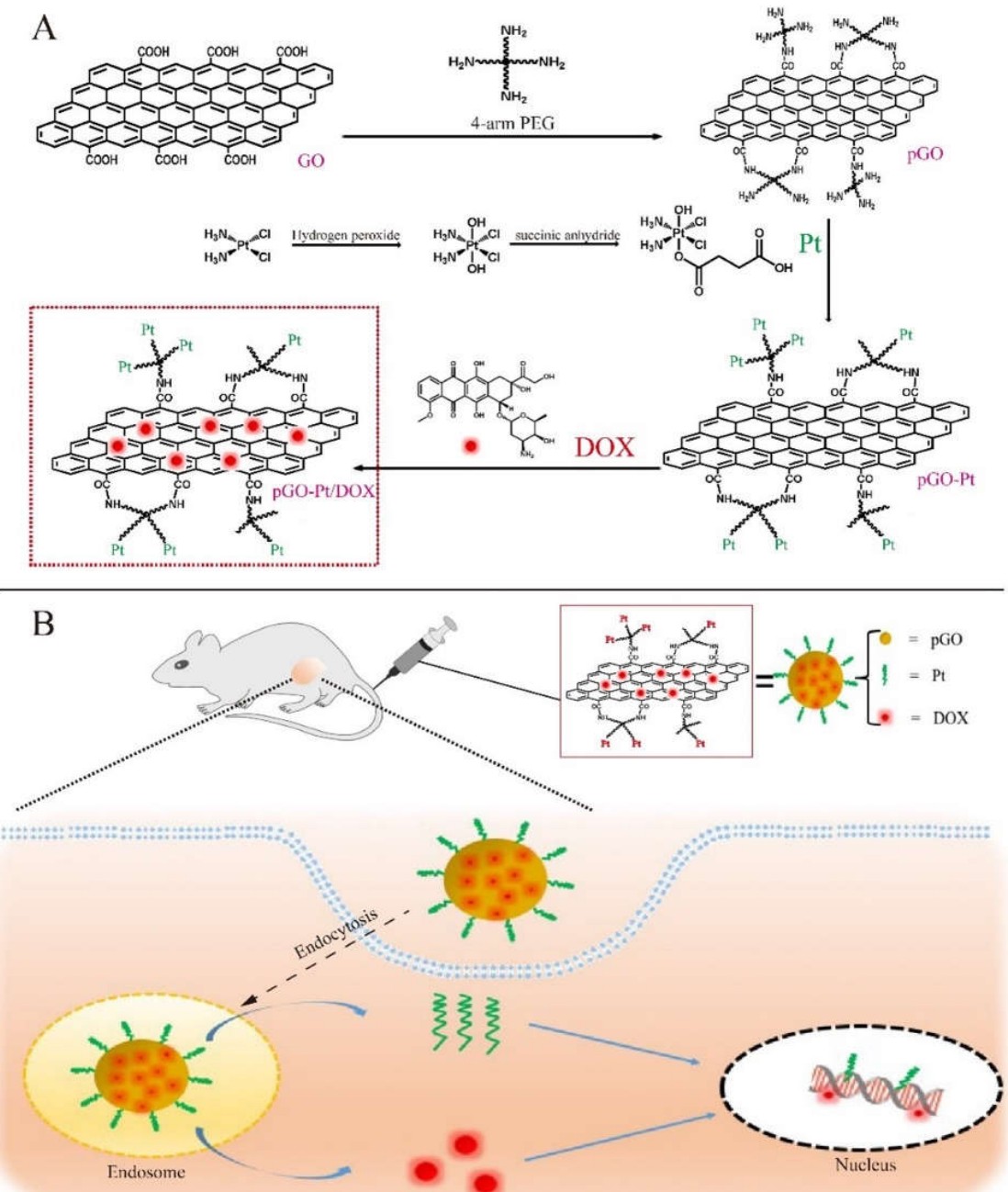

**Figure 5.** Illustration of synchronized codelivery of cisplatin (Pt) and doxorubicin (DOX) by polyethylene glycol-modified GO (PEGylated GO or pGO). (**A**) Steps in the synthesis of the Pt-pGO-DOX dual-drug delivery system, where covalent functionalization of GO with PEG created pGO, followed by covalent and noncovalent functionalization with Pt and DOX, respectively. (**B**) Schematic showing the mechanism of action of tumor targeting by the Pt-pGO-DOX drug delivery system, in which Pt generates DNA lesions and DOX inhibits DNA replication. (Reproduced with permission from [45]. Copyright 2020, Nature).

Zhang et al. described a system for targeted drug delivery using GO and FA ligands [92]. FA receptor-ligand binding is a well-studied Trojan-horse binding mechanism [93]. This mechanism involves the internalization of NPs and the release of a high number of toxic ions to induce cytotoxicity. GO was surface modified with sulfonic groups, which provided stability in physiological buffer solutions, and the hydroxyl groups present were converted into carboxyl groups for further interaction with FA. Then, the FA ligand was covalently immobilized onto the GO surface to obtain FA-GO, and the conjugate was studied for target-specific binding with the human

breast cancer cell line MCF-7 (FA receptor-positive). In the reported work, two anticancer drugs, DOX and CPT, were loaded in combination onto FA-GO through noncovalent binding via π-π interactions. The reported results demonstrated the efficiency of FA-GO with CTP and DOX to specifically interact with FA receptors on MCF-7 cells, and cellular uptake by endocytosis showed the stability of the conjugate. The cytotoxicity of the combined FA-GO/DOX/CPT conjugate was much higher than that of the conjugate loaded with a single drug. Thus, the system demonstrated the efficiency and viability of GO as a nanocarrier. Ye et al. reported the modification of GO with FA and β-cyclodextrin (CD) to achieve an improved carrying capacity and release controllability [94]. CD is a well-known and widely available cyclic oligosaccharide and is known for its capacity to hold drugs and increase solubility and slow release. In the reported article, CPT was loaded into the nanocarrier, and the release of the drug in phosphate-buffered saline (PBS) was studied against unmodified GO with CPT. GO-FA-CD exhibited superior control and slow release of CPT compared with GO loaded with CPT, demonstrating a modification and functionalization strategy for improving the control of drug release. Using the hydrophilic excipient 1-ethenylpyrrolidin-2-one, commonly known as polyvinylpyrrolidone (PVP), and CD to enhance stability and carrying capacity, Karki et al. modified GO nanocarriers with the drug SN38 to maximize cytotoxicity to cancer cells in vitro [95]. Controlled release of the drug was achieved in a pH-dependent manner, and SN38 loaded onto GO-PVP-CD showed significantly higher cytotoxicity than free SN38 in MCF-7 cells. Using this system in real diagnostics can be deleterious in the absence of a mechanism to specifically target malignant cells. The integration of cell receptor-specific ligands is required to achieve the expected efficacy.

For many decades, natural herbs have been reported to have many medicinal and therapeutic properties and activity against various diseases. One of these herbs is *Typhonium giganteum*, which is reported to inhibit tumor progression and restrict cellular growth. The presence of compounds such as β-sitosterol and lignin in the extracts of the herb has been thought to provide its anti-cancer properties. Gu et al. reported the modification of GO with biodegradable poly(lactic acid) (PLA) and poly(butylene carbonate) (PBC) to construct nanocarrier matrix GO-PLA-PBC fibers [96]. When lung carcinoma A549 cells were exposed to the nanofiber matrix GO-PLA-PBC loaded with ethanol extracts of the herb, cell shrinkage was observed followed by growth restriction, while control A549 cells incubated with GO-PLA-PBC exhibited continued growth. This work demonstrated the use of GO as a vehicle to carry and deliver the drug with specificity.

A few simpler strategies of delivering drug molecules have been reported that are not target-specific. Many reported studies have shown that DOX directly binds to the GO surface with a high loading capacity via π-π interactions and that release of the drug is pH-dependent [97]. This type of delivery system is highly efficient for targeting tumor cells because malignant cells are known to be slightly acidic, and in this environment, drug release is optimal. One novel and interesting strategy was developed by Wang et al., who combined gold NPs (AuNPs) with GO, and the conjugate was loaded with DOX molecules [98]. In the experiments, the DOX-AuNP-GO nanocarrier conjugate exhibited significantly higher cytotoxicity in HepG2 cells than DOX alone, demonstrating a higher carrying capacity and better transport. Yang et al. used iron(II, III) oxide ($Fe_3O_4$) NPs to bind with GO and DOX as drug delivery vehicles, which showed high toxicity in the human breast cancer cell line SK3 [99]. Drug release was shown to be pH-dependent, and drug uptake studies were performed using FITC-labeled GO (Figure 6). In a similar approach, Zheng et al. modified reduced GO (rGO) with $Fe_3O_4$ with magnetic properties to load the anticancer drug β-lapachone (β-lap) in an effort to provide efficient delivery to cells [73]. In that study, $Fe_3O_4$/rGO/β-lap showed remarkably higher cytotoxicity than $Fe_3O_4$/rGO in MCF-7 cells. The internalization of the nanoconjugates was tested using FITC-labeled $Fe_3O_4$/rGO. Fine-tuning of the described system with cell-specific targeting would be beneficial for in vivo and real-world drug delivery applications.

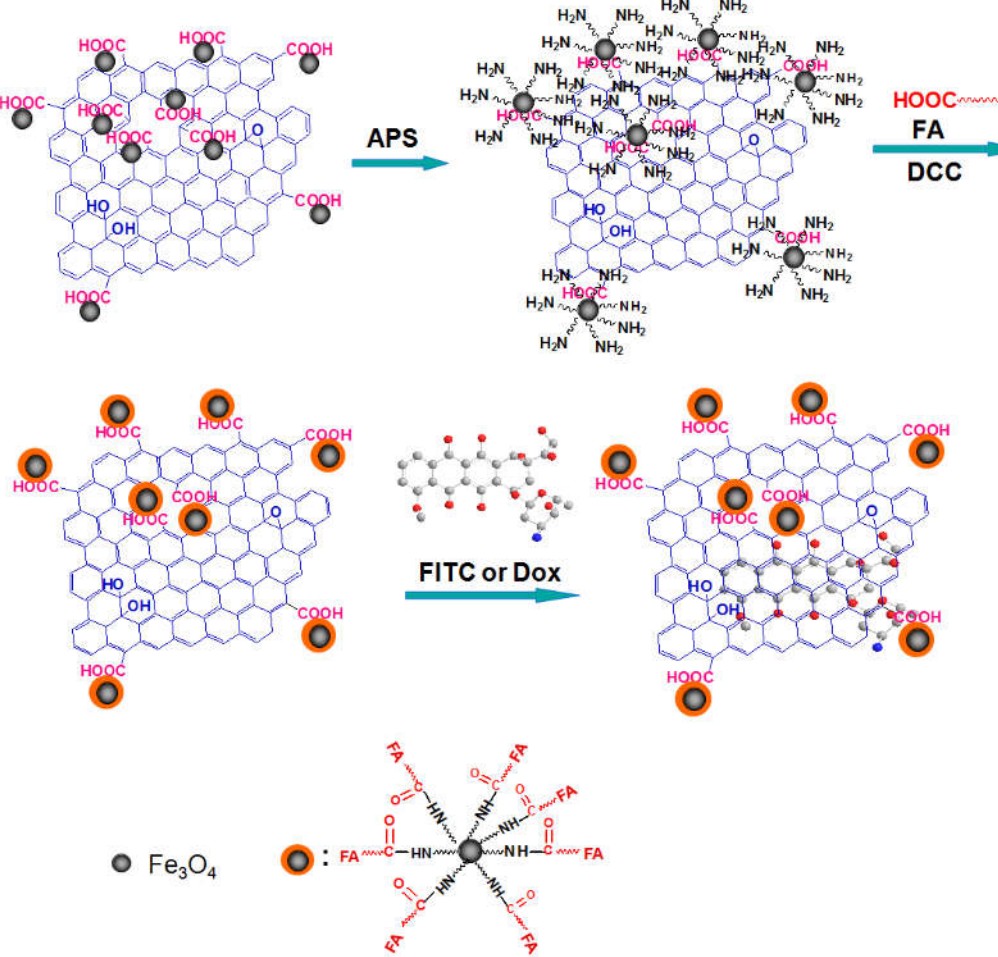

**Figure 6.** Synthesis of a multi functionalized drug delivery system based on GO and magnetite NPs. (Reproduced with permission from [99]. Copyright 2011, Royal Society of Chemistry).

3,3′-Diindolylmethane (DIM) is extracted from cruciferous plants and has been shown to exhibit anticancer effects [100]. The mechanism of its anticancer activity is still unknown. Recently, Deb and colleagues developed a graphene material-based nanocarrier system to successfully deliver DIM and CPT in vitro and in vivo [101]. Previously, this group of researchers discussed the potential differences and biocompatibility of graphene material modified with either PVP or CS, where CS showed higher biocompatibility [102]. Co-delivery of CPT and DIM was tested on CS-modified GO bionanocomposites in vivo in female Wistar rats.

*5.2. Use of GO-Based Nanocarriers in Photothermal and Photodynamic Therapy*

In contrast to conventional chemotherapy, photothermal and photodynamic therapy are relatively new approaches that are believed to be lethal to tumor cells and employ specific light irradiation with high specificity. Nanoscale GO shows excellent chemical and optical properties that are extremely useful in phototherapy. In this method, near-infrared (NIR) light of 700–100 nm is used to induce hyperthermia in tumor cells. GO exhibits light absorption in the NIR range, which efficiently increases the tissue temperature above 40 °C, destroying cells. Exploiting this property, Yang et al. reported that PEGylated graphene nanosheets (NGSs) exhibited high cellular uptake and potent photothermal properties useful for therapy under radiation from low-power NIR lamps [103]. The researchers further experimented with the size of the material to understand how it impacts the photothermal efficiency. NGS with a PEG coating exhibited enhanced NIR absorption and extremely lethal behavior, resulting in almost 100% elimination of tumor cells. In a similar study, Markovic et al. demonstrated a higher photothermal anticancer efficiency of PVP-coated graphene NPs (PVP-GNPs) than CNTs under similar experimental conditions [104]. It has been

discussed that the intensity of NIR radiation absorption by graphene is size-dependent, and the mechanism is not clearly understood [105]. Robinson et al. used PEGylated ultra-small rGO sheets functionalized with an RGD (Arg-Gly-Asp) peptide motif for selective targeting of U87MG brain cancer cells and achieved highly efficient photoablation of tumor cells [106]. Zhang et al. developed a combination treatment of DOX and photothermal therapy to treat tumor cells [107]. Nanoscale sheets of GO were modified with DOX in the presence of PEG, resulting in complete ablation of the tumor due to synergistic effects from DOX and the photothermal effects of GO, and no recurrence of the tumor was observed. This approach has been used and reported many times over the years, demonstrating the efficiency and efficacy of GO as a nanocarrier.

Photodynamic therapy relies on the ability of photosensitizer molecules to generate reactive oxygen species (ROS) to induce cell death under specific light irradiation. Dong et al. used PEGylated GO to attach zinc phthalocyanine (ZnPc), a well-known photosensitizer, on the surface of the nanocarrier [108]. The obtained conjugated ZnPc-PEG-GO particles showed high cytotoxicity against malignant cells under irradiation with Xe light. Hu et al. demonstrated the ability of a GO-titanium dioxide nanocomposite (GO-TiO$_2$) to exert a photodynamic effect within the visible light spectrum [109]. Huang et al. prepared a GO-FA nanoconjugate with good dispersity and low cytotoxicity to target malignant cells via photodynamic therapy [110]. GO nanocarriers were loaded with the second-generation photosensitizer chlorine e6 (Ce6) via hydrophobic and $\pi$-$\pi$ interactions. In the reported study, GO nanocarriers efficiently delivered and increased the concentration of Ce6 in MGC803 cells, leading to an excellent photodynamic effect upon irradiation.

### 5.3. Gene Delivery by GO-Based Nanocarriers

One of the most promising tools to treat complex genetic disorders is gene therapy, which has witnessed both breakthrough successes and tragic fatalities in clinical tests. Although the scope of germline gene therapy is controversial in terms of ethics and safety, gene therapeutic strategies and attempts involving bone marrow and blood are within legal grounds. The key to successful gene therapy is the efficiency of the vector in terms of protecting the genetic material from nucleases, restricting nonspecific binding, and ensuring cellular uptake. An efficient gene delivery vector with the abovementioned qualities is still lacking. Poly(iminoethylene), also known as polyethylenimine (PEI), is a cationic polymer known for its favorable binding affinity toward nucleic acids and a high degree of cellular uptake. In contrast, PEI exhibits significant toxicity and poor biocompatibility when used alone, restricting its potential application in therapeutics. To tackle this problem, researchers have conjugated positively charged PEI with GO to develop a nanocarrier. Chen et al. reported a novel gene delivery system using GO as a nanocarrier carrying reporter plasmids [111]. Branched PEI was grafted onto the GO surface to form PEI-GO, which exhibited efficiency in condensing genetic material on the GO surface and, in turn, delivered the genetic material, green fluorescent protein (pGFP)-expressing and luciferase-expressing (pGL-3) plasmids, to the nucleus of HeLa cells (Figure 7A). The delivery of the genetic material by the nanocarrier was confirmed by fluorescence microscopy. In a recent study, gene delivery was successfully achieved using magnetic NPs (mNPs) [112]. A GFP-encoding gene was delivered into MCF-7 cells using PEI-coated mNPs (pDNA-PI-mNPs). Magnetofection was achieved by placing culture plates over a magnetic field while cells were incubated with pDNA-PI-mNPs (Figure 7B). The abovementioned strategies successfully demonstrated the efficiency of gene delivery by biocompatible NPs. To date, a few lipid NP-based gene delivery solutions have been approved by the Food and Drug Administration (FDA), one of which is the siRNA drug Onpattro, developed by Alnylam, targeting hereditary amyloidosis [113]. In many reports over the years, GO has been demonstrated to be a promising tool for gene delivery. However, more analysis and biocompatibility studies are required to prepare a formulation for real-world drug delivery uses.

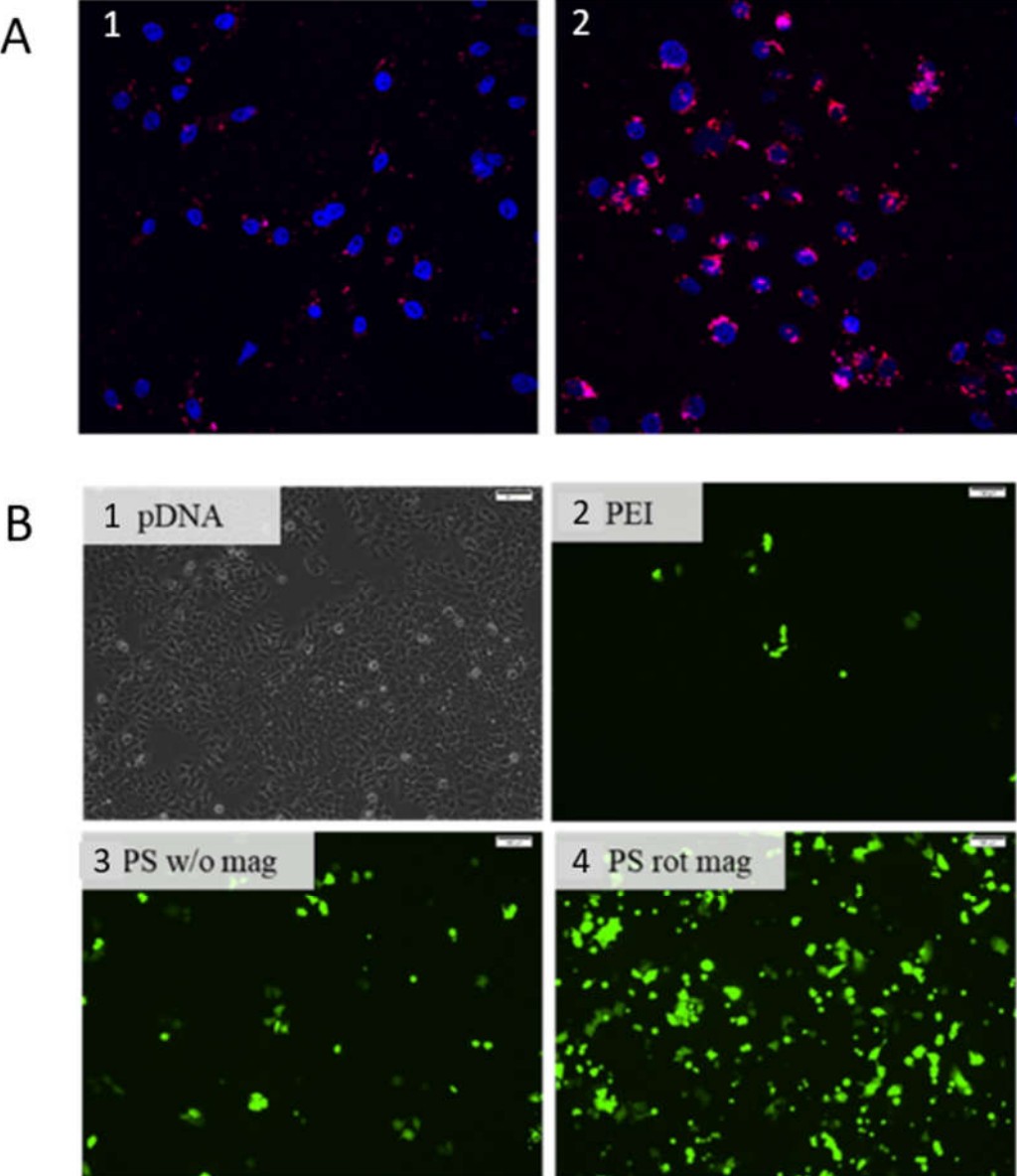

**Figure 7.** Intracellular imaging experiment tracking fluorescent particles delivered by NPs. (**A**) Polyethyl-enimine modified GO with Luciferase reporter vector (PEI-GO/pGL-3) complexes labeled with Cy3 (red) in HeLa cells at (**1**) 4 h and (**2**) 24 h post-transfection. 4′,6-diamidino-2-phenylindole (DAPI) staining was used to stain nuclei (blue). (Reproduced with permission from [111]. Copyright 2011, Royal Society of Chemistry). (**B**) Expression of green fluorescent protein (GFP)-encoding genes, delivered or not delivered by magnetic NPs, was observed with inverted fluorescence microscopy at 48 h post-infection (**1**) pDNA; (**2**) PEI; (**3**) PS w/o mag; (**4**) PS rot mag (Reproduced with permission from [112]. Copyright 2019, American Chemical Society (Permission for reuse was acquired directly from ACS. Any further permission for reuse or reproduction-related to the material excerpted should be directed to the ACS. Source of the material: https://pubs.acs.org/doi/10.1021/acsomega.9b01000)).

Nuclear localization signals (NLSs) are effective genetic tools to direct the entry of a foreign object into a cell across the semipermeable membrane. The PV7 peptide (PKKKRKV) is one such primary NLS. Ren et al. designed GO-PEI modified with PV7 to construct a nuclear-localized gene delivery system [114]. For the intercellular tracking study, GO-PEI-PV7 was labeled with Cy3. Successful uptake of the Cy3-labeled construct indicated the benefit of modifying GO-PEI with the PV7 NLS. GO-PEI-PV7 was also loaded with pEGFP (plasmid encoding enhanced green fluorescent protein) to show the delivery efficiency of the nanoconjugate.

*5.4. Delivery of Multifunctional Therapeutics by GO-Based Nanocarriers*

The efficacy reliance of conventional therapies on the potency of a single therapeutic agent is often limited owing to the large degree of complexity in cellular signaling and the defense mechanisms employed by cells in complex disorders. On the other hand, administration of a single drug at high concentrations to achieve the desired result often leads to severe toxicity. In combination therapy, two or more therapeutic agents can be loaded onto a single vector and administered to achieve the overall goal. The planar 2D structure of GO is advantageous because it results in a large surface area suitable for large carrying capacity and allows binding of two or more therapeutic agents via proper modification.

One simple approach to carrying two anticancer drugs, DOX and CPT, on a GO-FA nanoconjugate has already been described above. Zhang et al. developed a combination therapy strategy to first limit drug resistance and then exert the activity of tumor-suppressing drugs [115]. In that report, anti-Bcl-2 (B-cell lymphoma 2) siRNA was chosen. The Bcl-2 protein is known for its anti-apoptotic activity in many cancer types and confers multidrug resistance (MDR). Knocking down Bcl-2 protein via RNA interference can reduce the MDR of a tumor, making it susceptible to a tumor-suppressing drug. PEI-GO was used to load and co-deliver the Bcl-2-targeting siRNA and small molecule DOX to HeLa cells. Notably, Bcl-2 protein expression was downregulated by up to 52.6%, and the PEI-GO-DOX-Bcl-2 siRNA nanoconjugate exhibited a strong synergistic anticancer effect. Bao et al. reported a similar study in which CS was covalently bound onto GO to increase solubility and biocompatibility [116]. CS-GO was used to load CPT and pRL-CMV, a plasmid expressing *Renilla* luciferase protein. In the experiments, CS-GO-CPT exhibited significantly high cytotoxicity toward HepG2 and HeLa cells, demonstrating an anti-cancer effect. Besides, the nanocarrier successfully delivered the plasmid, which was confirmed by a *Renilla* luciferase assay.

Deb et al. designed a nanoconjugate of GO modified with CS and FA to co-load two anticancer drugs, CPT and 3,3′-diindolylmethane (DIM), to introduce a synergistic cytotoxic effect in vitro [101]. The two different mechanisms of action of the drugs made the conjugate highly toxic to breast cancer cells. In that study, the anticancer effect of free CPT was reported to be significantly lower than that of the GO-CS-FA-CPT-DIM conjugate in MCF-7 cells. In vivo histopathological studies were performed to analyze the biodistribution and bioavailability of the drugs. Additionally, in vivo experimental reports suggested that DIM masked the toxic effects generated by CPT.

Rasoulzadehzali et al. prepared hydrogel beads based on GO-silver NP clusters (GO-AgNPs) modified with CS and loaded DOX on the surface [117]. AgNPs exhibit antibacterial activity against both Gram-positive and Gram-negative bacteria [118]. In that study, GO-AgNP-CS-DOX conjugates showed a significant anticancer effect in human colon cancer cells (SW480) and exhibited an antibacterial effect against *Escherichia coli* and *Staphylococcus aureus.*

A comprehensive list of recently reported graphene material-based nanocarriers for various biochemical molecules has been presented in Table 3.

**Table 3.** Comprehensive list of GO-based nanocarriers for delivery of biochemical molecules.

| Type of Therapy | Therapeutic Agent | Nanocarrier Material | Application Study | Ref. |
|---|---|---|---|---|
| Drug molecule | DOX | NGO-PEG-rituxan-DOX | In vitro | [86] |
| | SN38 | NGO-PEG-SN38 | In vitro | [87] |
| | DOX | GO-DOX | In vitro | [88] |
| | DOX | pGO-Pt-DOX | In vitro, in vivo | [45] |
| | DOX | GO-CMC-DOX | In vitro | [91] |
| | CPT | GO-FA-CD-CPT | In vitro | [94] |
| | *Typhonium giganteum* extract | GO-PLA-PBC | In vitro | [96] |
| | DOX | GO-CS-FA-DOX | In vitro | [97] |
| | DOX | GO-AuNP-DOX | In vitro | [98] |
| | DOX | GO-Fe$_3$O$_4$-FA-DOX | In vitro | [99] |
| | DOX | GO-PVP-CD-DOX | In vitro | [95] |
| | DOX | GO-ZnO-DOX | In vitro | [85] |
| | Ellagic acid (EA) | GO-Pluronic F38, GO-Tween20, GO-maltodextrin | In vitro | [119] |
| | DOX | GO-CS-DOX | In vitro | [83] |
| | β-Lapachone | Fe$_3$O$_4$/rGO/β-lap | In vitro | [73] |
| | Paclitaxel (PTX) | GO-PEG-PTX | In vitro | [49] |
| | DOX | GO-PEG-DOX | In vitro | [50] |
| | CPT, DIM | GO-CS-FA-CPT-DIM | In vitro, in vivo | [101] |
| | Curcumin | GO-PEG-curcumin | In vitro | [51] |
| Photothermal, Photodynamic | NGS | NGS-PEG | In vivo | [103] |
| | GNP | GNP-PVP | In vivo | [104] |
| | rGO | rGO-PEG-RGD peptide | In vitro | [106] |

| | Chlorine e6 | GO-PEG-BPEI-Ce6 | In vitro | [120] |
|---|---|---|---|---|
| | ZnPc | GO-PEG-ZnPc | In vitro | [108] |
| | TiO₂ | GO-TiO₂ | In vitro | [109] |
| Gene | pGFP, pRL | GO-PEI-pGFP/pRL | In vitro | [111] |
| | pEGFP, pGL3 | GO-PEI-PV7-pEGFP/pGL3 | In vitro | [114] |
| | pCMV-Luc | GO-BPEI-pCMV Luc | In vitro | [121] |
| | Cy3, pEGFP | GO-PEI-PV7-pEGFP | In vitro | [114] |
| Combination | DOX, CPT | GO-FA-DOX-CPT | In vitro | [92] |
| | NGO, DOX | NGO-PEG-DOX | In vivo | [107] |
| | Anti-Bcl-2 siRNA, DOX | GO-PEI-siRNA-DOX | In vitro | [115] |
| | CPT, pRL-CMV | GO-CS-CPT-pRL | In vitro | [116] |
| | AgNP, DOX | GO-AgNP-CS-DOX | In vitro | [117] |
| | AgNP, simvastatin | GO-AgNP-simvastatin | In vitro | [122] |

## 6. Discussion and Future Perspective

The emergence of complex disorders and novel infectious diseases has led to a discussion within the scientific community regarding whether conventional drug therapy with its limitations is sufficient. As mentioned earlier, even with high in vitro efficiency, very few drugs achieve satisfactory in vivo therapeutic applications, especially due to challenges regarding the delivery system. Chemotherapeutic drugs efficiently kill most tumor cells, but often fail to reach the core of the tumor, resulting in recurrence of the malignancy. The magnitude of the molecular complexity within a cell and the drug resistance of diseases are the reasons behind the need for novel and efficient drug delivery systems that are biocompatible and avoid being harmful or toxic to surrounding tissues. In other words, drug delivery by a nanocarrier must be target-specific and able to control the release of the drug in a suitable and expected environment.

Graphene-based materials have shown true potential in drug delivery systems as biocompatible nanocarriers that can load a high amount of biochemical molecules, owing to their favorable structure. Although in vitro and in vivo studies have shown the potential of GO as a drug delivery nanocarrier in preclinical experimental setups, the use of these nanocarrier systems in real-world therapeutic applications is still far off because a few critical challenges remain. Surface modification and functionalization with polymers and other chemicals enhance the carrying capacity, and immobilization of target-specific ligands ensures delivery of the drug to the right receptor. Covalent and noncovalent bonding have successfully been used to increase the colloidal stability and biocompatibility essential for maintaining the nanocarrier structure in physiological buffers. Controlling the size and an exact number of GO layers synthesized from bulky graphite material is still challenging and essential because the physical properties of nanomaterials play an important role in successful cellular uptake. A deeper understanding of the synthesis method and required modifications are needed to reproducibly produce materials with similar dimensions and shapes.

Toxicity and cellular uptake are key issues in the efficacy of drug delivery mechanisms. Various recently concluded studies have reported GO as a nanocarrier that is biocompatible and causes low to insignificant toxicity, while there are also conflicting reports. The primary driving force to introduce and assimilate nanobiotechnology into medicine and real-time therapeutics are the possible positive impact on the economy. Nanotechnology has been used in the biomaterial field for the preparation and modification of materials, such as hard or soft tissue implants, antibiotic materials, and diagnostics [123]. The European Medicines Agency (EMA) is the current regulatory authority that assesses risks and supervises medicines in the European Union to protect and promote public (animal) health. According to the conventional regulatory framework within the EMA, the EMA currently supervises nanotherapeutic products containing nanomaterials. The legislation addresses specific products consisting of nanomaterials (e.g., cosmetic products, novel biocidal products, and medical devices), which also includes labeling and assessment of safety [124]. However, more studies, large screenings, and expert evaluations are required in the future to conclude the usage, safety, and quality of nanomaterials because of their complex behaviors. Deeper insight into the molecular interactions between cell surface receptors and GO is critical to understanding cellular uptake, which will, in turn, facilitate the development of better and more efficient drug delivery systems with GO nanocarriers. To conclude, GO has shown true potential as a nanocarrier in advanced therapeutic applications and should act as a building block for future nanomedicines and help in developing precise, target-specific, safe drug delivery systems.

**Author Contributions:** Investigation, S.M.; writing—original draft preparation, S.M.; writing—review and editing, Z.B. and A.A.; supervision, V.A., and L.R.; funding acquisition, L.R. All authors have read and agreed to the published version of the manuscript.

**Funding:** This work was supported by ERDF "Multidisciplinary research to increase application potential of nanomaterials in agricultural practice" (No. CZ.02.1.01/0.0/0.0/16_025/0007314) and by the Ministry of Education, Youth and Sports of the Czech Republic under the CEITEC 2020 project (LQ1601).

**Acknowledgments:** We would like to thank Debamudra Guha for technical assistance in formatting the tables in this article.

**Conflicts of Interest:** The authors declare no conflicts of interest.

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
