# Peer review of "Graphene Oxide as a Nanocarrier for Biochemical Molecules: Current Understanding and Trends"

_processes, doi:10.3390/pr8121636_

Round 1

Reviewer 1 Report

The review manuscript is interested to readers of drug delivery. However, the language needs to be revised by native speaker and more up-to date citations need to be added.

Author Response

Reviewer 1, comments:

The review manuscript is interested to readers of drug delivery. However, the language needs to be revised by native speaker and more up-to date citations need to be added.

Response: We thank the reviewer for his remark. The correction has been done accordingly to the reviewer’s comment. The manuscript was corrected by a native English speaker. Also as per reviewer’s suggestion, new up-to date citations have been added to the revised manuscript.

Reviewer 2 Report

The present manuscript describes and reviews the recent uses of graphene oxide (GO) as a nano-carrier in the design of efficient drug delivery systems. Concepts are very well explained and also accessible for the non-specialist in this field. Yet, also the challenges in the synthesis and design of such a nano-carrier drug delivery system are well pointed out and detailed. An outlook on the future prospects of GO-based drug delivery systems in the future is provided. I found this paper very interesting to read and would certainly recommend publication.

I only have one minor yet general issue: I would advise the authors to have their manuscript proof-read by a native English speaker. There are a number of recurring issues in grammar and expression, along with the use of wrong or missing words. Just to give two examples; line 42 Having said that ....; line 69 ... graphene-based ??? including ....

Author Response

Reviewer 2, comments:

The present manuscript describes and reviews the recent uses of graphene oxide (GO) as a nano-carrier in the design of efficient drug delivery systems. Concepts are very well explained and also accessible for the non-specialist in this field. Yet, also the challenges in the synthesis and design of such a nano-carrier drug delivery system are well pointed out and detailed. An outlook on the future prospects of GO-based drug delivery systems in the future is provided. I found this paper very interesting to read and would certainly recommend publication.

I only have one minor yet general issue: I would advise the authors to have their manuscript proof-read by a native English speaker. There are a number of recurring issues in grammar and expression, along with the use of wrong or missing words. Just to give two examples; line 42 Having said that ....; line 69 ... graphene-based ??? including ....

Response: We thank the reviewer for his remark. The correction has been done accordingly to the reviewer’s comment. The manuscript was corrected by a native English speaker. The revised manuscript has been read multiple times for the correction of issues with grammar or vocabulary.

line 42: Having said that ....

            We thank the reviewer for his comment. The part has been corrected.

line 69: ... graphene-based ??? including ....

            We thank the reviewer for his comment. The part has been corrected.

Reviewer 3 Report

The authors tryed to give an overview of the studies carried out about GO-based nanocarriers.

The review is well written anc comprehensive. It can be taken in consideration for the publication.

Author Response

Reviewer 3, comments:

The authors tryed to give an overview of the studies carried out about GO-based nanocarriers.

The review is well written anc comprehensive. It can be taken in consideration for the publication.

Response: We thank the reviewer for his comment.

Reviewer 4 Report

The focus of the review is to summarize the potenzialities of GO-based nanocarries and the improvementes for drug delivery. The review is well written I suggestet to expand the literature in the introduction, in particular where the authors describe that the physical properties of GO are dependent on the particular method of synthesis and degree of oxidation, at this point I suggest to quote this reference: Mugnano, M., Lama, G.C., Castaldo, R., Marchesano, V., Merola, F., del Giudice, D., Calabuig, A., Gentile, G., Ambrogi, V., Cerruti, P. and Memmolo, P., 2019. Cellular Uptake of Mildly Oxidized Nanographene for Drug-Delivery Applications. ACS Applied Nano Materials3(1), pp.428-439.

Author Response

Reviewer 4, comments:

The focus of the review is to summarize the potenzialities of GO-based nanocarries and the improvementes for drug delivery. The review is well written I suggestet to expand the literature in the introduction, in particular where the authors describe that the physical properties of GO are dependent on the particular method of synthesis and degree of oxidation, at this point I suggest to quote this reference: Mugnano, M., Lama, G.C., Castaldo, R., Marchesano, V., Merola, F., del Giudice, D., Calabuig, A., Gentile, G., Ambrogi, V., Cerruti, P. and Memmolo, P., 2019. Cellular Uptake of Mildly Oxidized Nanographene for Drug-Delivery Applications. ACS Applied Nano Materials, 3(1), pp.428-439.

Response: We thank the reviewer for his comment. The manuscript has been revised according to the reviewer’s comment.